# FCAUSE: FLOW-BASED CAUSAL DISCOVERY

## ABSTRACT

Current causal discovery methods either fail to scale, model only limited forms of functional relationships, or cannot handle missing values. This limits their reliability and applicability. We propose *FCause* a new flow-based causal discovery method that addresses these drawbacks. Our method is scalable to both high-dimensional as well as large volume of data, is able to model complex non-linear relationships between variables, and can perform causal discovery under partially observed data. Furthermore, our formulation generalizes existing continuous optimization-based causal discovery methods, providing a unified view of such models. We perform an extensive empirical evaluation, and show that *FCause* achieves state of the art results in several causal discovery benchmarks under different conditions reflecting real-world application needs.

## 1 INTRODUCTION

Understanding causal relationships between variables is crucial in many applications, including biology (Koller & Friedman, 2009; Sachs et al., 2005), economics (Varian, 2016; Cunningham, 2020), and healthcare (Tu et al., 2019). In addition, such information can also be used to advance other machine learning domains, such as fairness (Kusner et al., 2017; Chiappa, 2019), privacy (Tople et al., 2020; Chandrasekaran et al., 2021) and robustness (Arjovsky et al., 2019; Zhang et al., 2020; Zheng et al., 2021). In practice, however, we often do not have *a priori* knowledge of causal relationships. We can gain this knowledge experimentally, e.g., through randomized control trials (Hariton & Locascio, 2018), but experiments are sometimes cost-prohibitive or unethical. In such cases, *causal discovery*, the task of learning the causal relationships from existing data is essential. While causal discovery methods have been studied for decades, classical approaches tend to have scaling issues and rely on domain-specific assumptions.

Recent advances in causal discovery use continuous optimization methods to overcome the scalability issues in classical approaches (Zheng et al., 2018; 2020). They frame the combinatorial optimization problem of learning a directed acyclic graph (DAG) as a smooth optimization problem under constraints, allowing the use of efficient algorithms for continuous optimization. In addition to scalability advantages, this represents a first step in bridging the gap between deep learning and causal discovery. However, continuous optimization-based causal discovery remains experimental. These methods cannot handle datasets with missing values and suffer from simple distributional assumptions, making performance sensitive to changes in the dataset's scale (Reisach et al., 2021; Kaiser & Sipos, 2021). These limitations directly impact these methods' reliability and applicability.

In contrast, when the causal graph—the structure but not functional forms of causal relationships—is already known, causal-aware (Bayesian) deep learning methods have many application advantages. One recent framework, *Carefl* (Khemakhem et al., 2021), fits an autoregressive flow model using the variables' causal ordering for the autoregressive transformations. This yields a model that can be used to answer complex causal queries, such as counterfactuals. Its performance for end-to-end causal discovery, however, is limited by its reliance on classical causal discovery methods for inferring causal structures, the cost of fitting several flow-based models, and an inability to handle datasets with missing values.

In this work, we show how continuous optimization methods can be combined with flow-based models to address these deficiencies. We propose *FCause*, an efficient flow-based causal discovery method able to capture complex nonlinear relations between variables, robust to data scaling, and able to handle datasets with missing values (missing at random). *FCause* uses powerful flow-based

models to capture complex relationships between variables (Khemakhem et al., 2021), performs efficient causal discovery using ideas from continuous optimization-based causal discovery methods (Zheng et al., 2018), and handles partially-observed datasets using amortized approximate inference to estimate distributions over missing values (Kingma & Welling, 2013).

In section 6, we perform extensive empirical evaluations of *FCause*, and observe that it achieves state-of-the-art results on multiple causal discovery benchmarks, and that its performance is robust with respect to changes in the scaling of the data (e.g., standardized data or not). In addition, *FCause* remains competitive when used on datasets with 30% of missing values, even comparing against baselines used on the corresponding fully-observed dataset.

Finally, we present a unified formulation of several continuous optimization-based causal discovery algorithms based on flow-based models (Khemakhem et al., 2021). This unified perspective facilitates the development of general new techniques to improve these methods, and allows a simple comparison between methods, shedding light on the benefits and drawbacks of each one.

## 2    RELATED WORK

Approaches for causal discovery from observational data can be classified into three broad groups: constraint-based, score-based and functional causal models (Glymour et al., 2019). Constraint-based methods employ conditional independence tests to discover the underlying causal structure (Spirtes & Glymour, 1991; Spirtes et al., 2000). Score-based methods find a causal structure by optimizing a score function (Chickering & Meek, 2015; Chickering, 2020). Functional causal models represent each variable as a function of its direct causes and some noise term (Shimizu et al., 2006; Hoyer et al., 2008). Classical causal discovery methods in these groups often struggle to scale to high dimensions, as the combinatorial space of possible causal orderings among variables grows super-exponentially (Peters et al., 2017). Thus, scalable causal discovery algorithms tend to reduce the search space through reliance on domain-specific assumptions.

Recently, Zheng et al. (2018) introduced *Notears*, a new family of score-based method based on continuous optimization. *Notears* uses a novel algebraic characterization of directed acyclic graphs (DAG), which allows an equality-constrained optimization problem to jointly learn the model parameters and adjacency relationships between variables. *Notears* inspired the development of other methods, *Notears-MLP* and *Notears-Sob* (Zheng et al., 2020), *Grandag* (Lachapelle et al., 2019), and *DAG-GNN* (Yu et al., 2019), which extend the original formulation to model nonlinear relationships between variables. The methods' main benefits are its scalability and simplicity, a consequence of the fact that standard numerical solvers can be used to solve the resulting optimization problem. However, they have often been observed to be sensitive to different data scalings (Kaiser & Sipos, 2021), and cannot handle missing values.

Additionally, normalizing flows (Rezende & Mohamed, 2015) have been used to build causal aware models (Cai et al., 2018; Khemakhem et al., 2021). These are based on the fact that the variables' causal ordering can be used for the transformations used by autoregressive flows (Kingma et al., 2016; Huang et al., 2018).[1] These methods are able to model complex nonlinear relationships between variables. However, they rely on prior traditional methods or local search algorithms for causal discovery. Specifically, *Self* (Cai et al., 2018) uses a hill climbing procedure, and *Carefl* (Khemakhem et al., 2021) proposes to use a constraint-based method (e.g. PC) to find the graphs skeleton with as many oriented edges as possible and to fit several flow models to determine the orientation of the remaining edges. Apart from aforementioned methods, causal aware deep learning models have, in general, shown better properties regarding generalization and robustness (Arjovsky et al., 2019; Zhang et al., 2020; Kyono et al., 2020; Tople et al., 2020).

## 3    PRELIMINARIES

In this section, we describe the representations of causal relationships (SEM and DAGs) and explain how causal discovery can be formulated as an optimization task. We bring these components together when we introduce *FCause* in section 4.

---

[1] While the formal connection to a restricted type of autoregressive flows was proposed by (Khemakhem et al., 2021), some previous methods (Mooij et al., 2011; Cai et al., 2018) use closely related ideas.

**Causal graphical models** (Peters et al., 2017) are widely used to represent causal relationships between variables. Each variable is assigned a node in a graph (typically a DAG), and causal relationships are represented as directed edges between nodes. These models are encoded using the graph's adjacency matrix $A \in \{0,1\}^{d \times d}$, where $d$ is the number of variables, or by explicitly stating the set of parents for each variables. If each of the edges is additionally assigned a weight, a weighted adjacency matrix is used $W \in \mathbb{R}^{d \times d}$ instead. Here, zero entries indicate the absence of an edge, and non-zero entries indicate the presence of an edge with the corresponding weight.

**Structural Equation Models (SEM)** are commonly used to describe a causal system. They characterize the value of a variable as a function of its causal parents and some external noise. Let $x = [x_1, \ldots, x_d]$ represent $d$ causally related variables, $A \in \{0,1\}^{d \times d}$ the binary adjacency matrix between variables, and $z = [z_1, \ldots, z_d]$ pairwise independent random noise. SEMs model each variable as $x_i = f_i(x_{\mathrm{pa}(i)}, z_i)$, where $f_i$ a scalar function and $x_{\mathrm{pa}(i)} = \{x_j : A_{j,i} = 1\}$ represents the set of parents of $x_i$ according to $A$.

**Additive Noise Models (ANM)** represent a specific type of SEM for which the noise is additive:

$$x_i = f_i(x_{\mathrm{pa}(i)}) + z_i, \qquad i = 1, \ldots, d. \tag{1}$$

Equivalently, given a fixed adjacency matrix $A$, these $d$ equations can be expressed jointly as

$$x = f_A(x) + z, \tag{2}$$

where $f_A(x)$ outputs a $d$-dimensional vector whose $i$-th component is given by $f_i(x_{\mathrm{pa}(i)})$. This is one of the most common functional forms in causal discovery. A crucial question regarding ANMs involves the identifiability of the underlying causal graph under observational data. Can the true adjacency matrix be recovered in the limit of infinite data? Hoyer et al. (2008) showed that it is possible except when the underlying system is a combination of linear functions with Gaussian noise. In this work, we assume that the underlying functions are nonlinear without any specific assumption on their functional form. In this setting, the causal graph can be identified.

**A new algebraic characterization of DAGs** was recently proposed by Zheng et al. (2018). They showed that, given a weighted adjacency matrix $W \in \mathbb{R}^{d \times d}$, the quantity

$$h(W) = \mathrm{tr}\left(e^{W \odot W}\right) - d \tag{3}$$

is non-negative, and zero if and only if $W$ represents a DAG. Since its introduction, this algebraic characterization of DAGs has been widely used to frame causal discovery problems with ANMs as continuous optimization tasks (Zheng et al., 2018; 2020; Lachapelle et al., 2019). All these methods propose to train a model's parameters $\theta$ by maximizing a score subject to the constraint $h(W(\theta)) = 0$, where $W(\theta)$ represents the weighted adjacency matrix as a function of the parameters $\theta$. Due to the non-convexity of the set $\{\theta : h(W(\theta)) = 0\}$, these methods use $h(W(\theta))$ as a penalty term added to the loss, whose weight is increased as optimization proceeds.

## 4  FCAUSE: FLOW-BASED CAUSAL DISCOVERY

We present *FCause*, a novel flow-based causal discovery method for nonlinear additive noise models. Our method learns a binary adjacency matrix jointly with the model parameters, is able to capture complex nonlinear relationships between variables, is stable to data re-scalings, and can handle datasets with missing values. We present the method for fully-observed datasets in section 4.1, and its extension to datasets with missing values in section 4.2. For the latter we assume that values are missing completely at random or missing at random, meaning that the missingness pattern either has no cause or has fully observed causes, which is a common setting (Rubin, 1976; Stekhoven & Bühlmann, 2012; Ma et al., 2018; Mattei & Frellsen, 2019; Strobl et al., 2018).

### 4.1  FCAUSE WITH FULLY-OBSERVED DATA

*FCause* takes a Bayesian approach to causal discovery (Heckerman et al., 1999). We model the adjacency matrix jointly with the observations as

$$p_\theta(x^1, \ldots, x^N, A) = p(A) \prod_n p_\theta(x^n | A). \tag{4}$$

We propose to fit the model parameters $\theta$, and to use the posterior distribution over binary adjacency matrices $p_\theta(A|x^1, \ldots, x^N)$ to characterize the causal structure learned by the model.

There are two challenges to applying this approach: (i) The posterior distribution over $A$ is intractable, and (ii) maximum likelihood cannot be used to fit the model parameters, due the the presence of the latent variable $A$. We overcome these using variational inference (Jordan et al., 1999; Blei et al., 2017; Zhang et al., 2018). We define a variational distribution $q_\phi(A)$ to approximate the intractable posterior $p_\theta(A|x^1, \ldots, x^N)$, and use it to build the evidence lower bound (ELBO), a lower bound on the marginal likelihood that can be used as a surrogate objective. This yields

$$\text{ELBO}(\theta, \phi) = \mathbb{E}_{q_\phi(A)} \left[ \log p(A) \prod_n p_\theta(x^n|A) \right] + H(q_\phi) \leq \log p_\theta(x^1, \ldots, x^N), \tag{5}$$

where $H(q_\phi)$ represents the entropy of the distribution $q_\phi$. (Details for the derivation in appendix B.)

**Modeling and optimization details** Following *Carefl*, we set $p_\theta(x^n|A)$ to an autoregressive flow with base distribution $p_z$ and transformation $z = g_A(x) = x - f_A(x)$ (cf. equations 1 and 2), and $p(A)$ to some prior over adjacency matrices that places zero mass on matrices that do not represent DAGs (needed to guarantee invertibility of the transformation, cf. appendix E). We get *FCause*'s final objective by replacing $p_\theta(x^n|A)$ in eq. (5) using the change of variable formula from random variables (Kingma et al., 2016). This yields

$$\text{ELBO}(\theta, \phi) = \mathbb{E}_{q_\phi(A)} \left[ \log p(A) \prod_n p_z \left( x^n - f_A(x^n; \theta) \right) |\det J_A(x^n; \theta)| \right] + H(q_\phi), \tag{6}$$

where $J_A(x; \theta)$ is the Jacobian of the transformation used, $g_A(x; \theta) = x - f_A(x; \theta)$. We set the base noise distribution $p_z$ to be a factorized Gaussian with mean zero and learnable variances, the variational distribution $q_\phi(A)$ to be the product of independent Bernoulli distributions, one for each entry in the matrix $A$,[2] and use the Gumbel-softmax to get stochastic estimates of the ELBO's gradient (Maddison et al., 2016; Jang et al., 2016).[3]

Interestingly, the objective from eq. (6) may be simplified by noting that the Jacobian-determinant term is always one for matrices $A$ that represent a DAG (see e.g. (Mooij et al., 2011). We formalize this in lemma 1 and include a proof in appendix D). Thus, since the prior over $A$ only allows DAGs, this term does not need to be computed.

**Lemma 1.** *Let $A$ represent a binary adjacency matrix, $f_A : \mathbb{R}^d \to \mathbb{R}^d$ a function whose $i$-th output only depends on the parents of $x_i$, and $J_A(x)$ the Jacobian of $g_A(x) = x - f_A(x)$. If the adjacency matrix $A$ represents a DAG, then $|\det J_A(x)| = 1$.*

The choice for $f_A : \mathbb{R}^d \to \mathbb{R}^d$ must satisfy the adjacency relations specified by $A$. That is, if $A_{j,i} = 0$, then the function $f_i(x)$ (i.e., the $i$-th component of the output of $f_A(x)$) must satisfy $\partial f_i(x)/\partial x_j = 0$. Inspired by Graph Neural Networks (Hamilton, 2020), we propose a flexible parameterization that satisfies this by setting

$$f_i(x) = h_i \left( \sum_{j=1}^d A_{j,i} \; g_j(x_j) \right), \tag{7}$$

where $g_i$ and $h_i$ ($i = 1, \ldots, d$) are multi-layer perceptrons. A naive implementation of this requires training $2d$ neural networks. To avoid this, we propose to parameterize these functions as $h_i(\cdot) = h(u_i, \cdot)$ and $g_i(\cdot) = g(u_i, \cdot)$, where $u_i$ is an $d$-dimensional trainable embedding. This simple idea reduces the number of MLPs needed from $2d$ to just 2.

Finally, as mentioned above, the prior $p(A)$ must assign zero probability to matrices that do not represent a DAG. While such a prior can be obtained in practice, it leads to a poorly conditioned

---

[2]We use the parameterization proposed by Lippe et al. (2021), which uses two parameters per each edge, one to model the existence and the other for the orientation.

[3]The use of the Gumbel-softmax to learn binary a adjacency matrix was also previously used by Ng et al. (2019a) and Brouillard et al. (2020).

optimization problem. Therefore, following Zheng et al. (2018), we use a "soft" alternative instead and gradually anneal it towards the "hard" DAG-forcing prior during training. This soft prior is

$$p(A) \propto \exp\left(-\lambda_s \|\text{vec}(A)\|_1 - \rho\, h(A)^2 - \alpha\, h(A)\right), \tag{8}$$

where $h(\cdot)$ is the non-DAG penalty from eq. (3) ($h(A) > 0$ for non DAGs and exactly zero for DAGs). The first term favours sparsity in the adjacency matrix $A$, and the latter two favour DAGs. We anneal this soft prior by increasing the weights $\rho$ and $\alpha$ as optimization proceeds. We include details on the full optimization procedure used in appendix A.

Note that the use of this soft prior means that only in the latter stages of training does the soft prior approach the hard DAG-enforcing constraint. While this means that in the initial training stages the assumptions of lemma 1 do not hold, we find that in latter stages all sampled matrices $A$ are DAGs, satisfying lemma 1's assumptions and providing strong empirical performance.

## 4.2 FCAUSE WITH PARTIALLY-OBSERVED DATA

We introduce *FCause* for partially-observed datasets. Missing values are naturally present in many domains (e.g. online education (Wang et al., 2021)). Thus, developing methods to handle them effectively is crucial. Traditional deletion-based methods fail in many applications since, after deletion, very few data points are left. In contrast, *FCause* handles missing values efficiently, by using approximate inference to estimate distribution over missing values.

We use $x_o^n$ to denote the observed components of sample $x^n$, $x_u^n$ to denote the unobserved components, and $p_\theta(x_o^n, x_u^n)$ to denote the probability of the sample obtained by combining the values in $x_o^n$ and $x_u^n$. As in the formulation without missing values, exact maximum likelihood is intractable. Thus, we propose to introduce a distribution to approximate the true posterior over the latent variables and to optimize the ELBO instead. However, in this case the latent variables include not only the adjacency matrix but also all unobserved values in the dataset. We define the variational approximation as

$$q_{\phi,\psi}(A, x_u^1, \dots, x_u^N | x_o^1, \dots, x_o^N) = q_\phi(A) \prod_n q_{\psi_n}(x_u^n | x_o^n), \tag{9}$$

which leads to the objective

$$\text{ELBO}(\theta, \phi, \psi) = \mathbb{E}_{q_{\phi,\psi}}\left[\log p(A) \prod_n p_\theta(x_o^n, x_u^n | A)\right] + H(q_\phi) + \sum_n H(q_{\psi_n}). \tag{10}$$

This is the objective maximized by *FCause* when the data has missing values. As explained before, we compute $p(x_o^n, x_u^n | A)$ via the change of variable formula for random variables, and set $q_\phi(A)$ to be the product of independent Bernoulli distributions. Finally, we set $q_{\psi_n}(x_u^n | x_o^n)$ to be a factorized Gaussian, and use reparameterization to get unbiased gradients with respect to $\psi_n$.

For efficiency, we use an amortization network (Kingma & Welling, 2013) to avoid training $N$ independent Gaussian approximations. To do so, we train a single neural network with parameters $\psi$, which receives as input the zero imputed sample $x^i$ concatenated with its "missingness" mask—which indicates indices of components missing for that sample—and outputs the mean and scale of $q(x_u^i | x_o^i)$. This allows us to train a single set of parameters $\psi$, which are shared across samples in the dataset, instead of $N$ set of independent parameters, $\{\psi_1, \dots, \psi_N\}$, one for each sample.

## 5 UNIFIED FLOW-BASED FORMULATION AND ANALYSIS

This section introduces a simple analysis showing that, similarly to *FCause*, most causal discovery methods based on continuous optimization can be framed from a probabilistic perspective as fitting a flow. The benefits of this unified perspective are twofold. First, it allows a simple comparison between methods, and sheds light on the different assumptions used by each one, their benefits and drawbacks. Second, it simplifies the development of new tools to improve these methods, since any improvements to one of them can be easily mapped to the others by framing them in this unified framework (e.g. our extension to handle missing values can be easily integrated with *Notears*).

The connection between causal discovery methods based on continuous optimization and flow-based models uses the concept of a weighted adjacency matrix $W(\theta) \in \mathbb{R}^{d \times d}$ linked to a function $f(x; \theta) : \mathbb{R}^d \to \mathbb{R}^d$. Loosely speaking, these matrices can be seen as characterizing how likely is each output of $f(x; \theta)$ to depend on each component of the input $x$. For instance, $W(\theta)_{j,i} = 0$ indicates that $f_i(x; \theta)$ is completely independent of $x_j$. Such adjacency matrices can be constructed efficiently for a wide range of parameterizations for $f$, such as multi layer perceptrons and weighted combinations of nonlinear functions (Zheng et al., 2018; 2020). We give precise details on their definition and construction in appendix C.

**Lemma 2.** *Let $f(x; \theta) : \mathbb{R}^d \to \mathbb{R}^d$ be a $\theta$-parameterized function with weighted adjacency matrix $W(\theta) \in \mathbb{R}^{d \times d}$. Given a dataset $\{x^1, \ldots, x^N\}$, fitting a flow with the transformation $z = x - f(x; \theta)$, base distribution $p_z$ and a hard acyclicity constraint on $W(\theta)$ is equivalent to solving*

$$\max_\theta \sum_n \log p_z(x - f(x; \theta)) \qquad \text{s.t.} \qquad h(W(\theta)) = 0, \tag{11}$$

*where $h(\cdot)$ is the algebraic characterization of DAGs from eq. (3).*

*Proof.* The acyclicity constraint is enforced by constraining the optimization domain to $\Theta = \{\theta : h(W(\theta)) = 0\}$. Then, the maximum likelihood objective can be written as

$$\sum_n \log p_\theta(x^n) = \sum_n \log p_z(x^n - f(x^n; \theta)) + \log \left| \det \frac{\mathrm{d}(x^n - f(x^n; \theta))}{\mathrm{d}x^n} \right| \tag{12}$$

$$= \sum_n \log p_z(x^n - f(x^n; \theta)) \qquad \text{(lemma 1),} \tag{13}$$

where the first equality we use the change of variable formula, valid because the transformation $z = g(x; \theta) = x - f(x; \theta)$ is invertible for any $\theta \in \Theta$. $\qquad \square$

Lemma 2 is the main building block in the formulation of continuous optimization-based causal discovery methods from a probabilistic perspective as fitting flow models. This is simply because the objective used by each of the methods can be exactly recovered from eq. (11) with specific choices for $f(x; \theta)$ and $p_z$.

***Notears* (Zheng et al., 2018)** uses a standard Gaussian for $p_z$ and a linear transformation for $f(x, \theta)$. (See appendix C for more details.)

***Notears-MLP* (Zheng et al., 2020)** uses a standard Gaussian for $p_z$ and $d$ independent multi-layer perceptrons, one for each component of $f(x, \theta)$.

***Notears-Sob* (Zheng et al., 2020)** uses a standard Gaussian for $p_z$ and a weighted linear combination of nonlinear basis functions.

***GAE* (Ng et al., 2019b)** uses a standard Gaussian for $p_z$ and a GNN for $f(x, \theta)$.

***Grandag* (Lachapelle et al., 2019)** uses a factorized Gaussian with mean zero and learnable scales for $p_z$ and $d$ independent multi layer perceptrons, one for each component of $f(x, \theta)$.

***Golem* (Ng et al., 2020).** This is a linear method whose original formulation was already in a probabilistic perspective, using the linear transformation for $f(x; \theta)$.

In summary, recently proposed causal discovery methods based on continuous optimization can be formulated from a probabilistic perspective as fitting a flow with different constraints, transformations, and base distributions. From these three components, the base distribution used by the flow is sometimes overlooked but plays an important role. It has been observed, both theoretically and empirically, that methods that use a standard Gaussian (e.g. *Notears*, *Notears-MLP*, *Notears-Sob*) may fail to recover the true underlying causal graph in simple cases, and that their performance may change significantly when re-scaling variables in the dataset (e.g. standardizing data or not) (Loh & Bühlmann, 2014; Kaiser & Sipos, 2021; Reisach et al., 2021). These issues can be overcome by setting the flow's base distribution to the actual noise distribution used to generate the data. While this distribution is typically unknown, methods such as *FCause*, *Grandag* and *Golem* take a step in this direction by optimizing some of its parameters, instead of fixing all of them to arbitrary values.

Finally, as mentioned above, this unified formulation simplifies the development of new tools to improve these methods. For instance, in this work we introduce a method to handle datasets with

missing values for *FCause*. Turns out that this idea is readily applicable to any of the causal discovery methods described in this section. This can be seen by formulating them as probabilistic models $p_\theta(x)$, and replacing the likelihood objective used for fully observed datasets with the ELBO from eq. (10) (and possibly removing the random variable $A$ from the objective).

# 6 EXPERIMENTS

This section presents results that empirically validate *FCause* for causal discovery. All simulations were performed using NVIDIA V100 GPUs.

**Datasets.** We consider synthetic, pseudo-real, and real data. For the synthetic data we follow the approach from Lachapelle et al. (2019) and Zheng et al. (2020). We sample a DAG following two different random graph models, Erdos-Renyi (ER) and scale-free (SF). For each graph we simulate $x_i = f_i(x_{\text{pa}(i)}) + z_i$, where $f_i$ is sampled from a Gaussian process and $z_i \sim \mathcal{N}(0, \sigma_i^2)$, with $\sigma_i \sim \mathcal{U}(0.2, 1)$. We consider three different dimensionalities $d \in \{16, 32, 64\}$, and for each $d$ we consider two possible number of edges $e \in \{d, 4d\}$. Each resulting dataset is identified as $\text{ER}(d, e)$ or $\text{SF}(d, e)$. All datasets have $n = 1000$ training samples. (In appendix F we show results in the exact same setting but with higher levels of noise, achieved by sampling $\sigma_i \sim \mathcal{U}(0.2, 2)$.)

For the pseudo-real dataset we consider data generated with the SynTReN generator (Van den Bulcke et al., 2006), which creates synthetic transcriptional regulatory networks and produces simulated gene expression data that approximates experimental data. We use the datasets generated by Lachapelle et al. (2019) (dimension 20), and take $n = 400$ for training. Finally, for the real dataset we use the widely used dataset that measures the level of several proteins in human cells from Sachs et al. (2005). We use a training set with $n = 800$ observational samples of dimension $d = 11$.

**Baselines.** We compare against *PC* (Kalisch & Bühlman, 2007), (linear) *Notears* (Zheng et al., 2018), the nonlinear variants *Notears-MLP* and *Notears-Sob* (Zheng et al., 2020), *Grandag* (Lachapelle et al., 2019) (without the preliminary neighborhood search), *GES* (Chickering, 2002), and *ICALiNGAM* (Shimizu et al., 2006). *GES* was obtained from the GES package, and all other baselines were obtained from the gcastle package.

**Causality metrics used.** Following common practice (Glymour et al., 2019; Tu et al., 2019) we report adjacency and orientation metrics (recall, precision, and F1 score), and causal accuracy (Claassen & Heskes, 2012). For *FCause*, which returns a distribution over adjacency matrices, we report the expected values of these metrics estimated using samples $A \sim q_\phi(A)$.

| | Avg Rank (lower is better) Based on | | |
|---|---|---|---|
| | Adjacency F1 | Orientation F1 | Causal Accuracy |
| *FCause* | **1.4 ± 0.7** | **1.3 ± 1.0** | **1.4 ± 1.3** |
| *Notears-MLP* | 3.5 ± 1.4 | 3.6 ± 1.3 | 4.3 ± 0.8 |
| *Notears-Sob* | 4.7 ± 0.7 | 3.2 ± 0.9 | 4.0 ± 0.7 |
| *GES* | 2.6 ± 0.8 | 3.4 ± 1.3 | 2.1 ± 0.5 |
| *PC* | 3.0 ± 1.2 | 4.4 ± 1.6 | 3.4 ± 1.0 |
| *Grandag* | 7.4 ± 0.8 | 6.5 ± 1.2 | 6.8 ± 1.0 |
| *ICALiNGAM* | 6.4 ± 1.0 | 6.4 ± 1.6 | 6.8 ± 0.8 |
| *Notears* | 6.9 ± 0.8 | 7.2 ± 1.1 | 7.3 ± 0.6 |

Table 1: ***FCause* ranks highest among all baselines.** Best results in bold. The table shows the methods average rank across datasets for different metrics (lower is better, 1 is the best possible, and 8 is the worst).

In all cases we standardize data and repeat all simulations for four different random seeds. (For the synthetic and pseudo-real datasets this includes generating four datasets, one for each seed.) Table 1 presents a high level summary of the results, by ranking the methods according to different metrics. It can be observed that *FCause* ranks highest, followed by *Notears-MLP*, *Notears-Sob*, *GES* and *PC* (all four perform similarly), and finally followed by *Grandag*, *ICALiNGAM* and *Notears*.

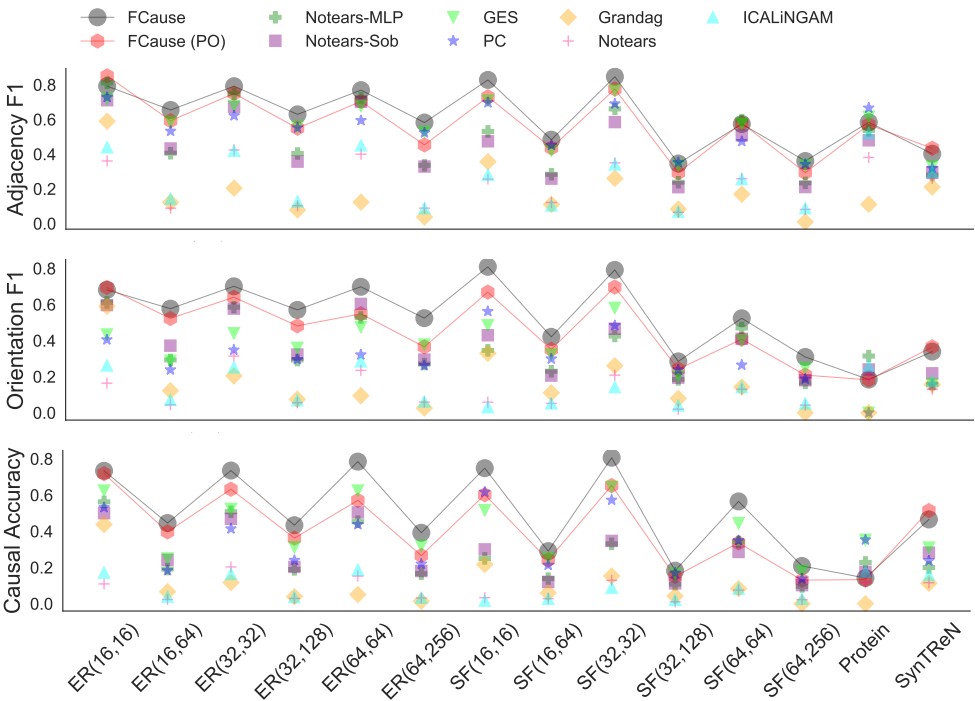

Figure 1: *FCause* **achieves better results than the baselines in all three metrics shown.** The legend "FCause (PO)" corresponds to running *FCause* with 30% of the training data missing completely at random. For readability, we highlight the *FCause* and *FCause (PO)* results by connecting them with soft lines. The figure shows mean results across four different random seeds. Results including standard deviations are included in appendix G.

Fig. 1 presents more detailed results. It shows the adjacency F1 score, orientation F1 score, and causal accuracy achieved by each method in each of the datasets considered. (We report full results, including standard deviations and all remaining metrics —precision and recall metrics for adjacency and orientation— in appendix G). Again, it can be observed that *FCause* tends to perform best across most datasets, followed by *Notears-MLP*, *Notears-Sob*, *GES* and *PC*.

The figure also shows the results achieved by *FCause* ran on the exact same datasets with 30% of the values missing. This is identified as *FCause (PO)*. As expected, the method performance gets slightly worse in the presence of missing values. However, it can be observed that it is still very competitive, and that on many datasets its performance is better than that of the baselines, even when these are ran using the dataset with no missing values. This shows the effectiveness of *FCause*'s way of dealing with missing values.

Results using datasets generated with a higher level of noise in are shown in fig. 2 (appendix F). As expected, as a consequence of the higher level of noise, all methods perform slightly worse. However, the main conclusions remain unchanged.

**Stability of FCause with respect to data scaling** Since *FCause* learns the scales of the base distribution, we expect its performance to be robust to different re-scaling of the variables in the dataset (cf. section 5). This is in contrast to other methods, such as *Notears-MLP* and *Notears-Sob*, who fix the base distribution to a standard Gaussian, and whose performance has been observed to change when re-scaling variables (Kaiser & Sipos, 2021; Reisach et al., 2021). We consider a simple experiment to verify this empirically. We ran *FCause*, *Notears-MLP* and *Notears-Sob* on all datasets with and without data standardization, and compare the results obtained in both cases.

Table 2 shows the performance achieved by each method when ran with and without data standardization. It can be observed that *FCause*'s mean performance is essentially the same in both cases.

| | Standard data | Adjacency | | | Orientation | | | Causal Accuracy |
|---|---|---|---|---|---|---|---|---|
| | | Recall | Prec | F1 | Recall | Prec | F1 | |
| *FCause* | True | 0.57 | 0.77 | 0.61 | 0.49 | 0.65 | 0.53 | 0.50 |
| | False | 0.58 | 0.76 | 0.62 | 0.50 | 0.68 | 0.55 | 0.51 |
| *Notears-MLP* | True | 0.37 | 0.85 | 0.49 | 0.27 | 0.62 | 0.35 | 0.27 |
| | False | 0.41 | 0.80 | 0.45 | 0.31 | 0.69 | 0.37 | 0.31 |
| *Notears-Sob* | True | 0.34 | 0.85 | 0.44 | 0.27 | 0.71 | 0.37 | 0.27 |
| | False | 0.49 | 0.51 | 0.42 | 0.38 | 0.45 | 0.35 | 0.38 |

Table 2: ***FCause* performance is robust with respect to data standardization.** The table shows the mean performance across datasets achieved by each method with and without data standardization.

| | Metric used to compute relative difference | | | | | | |
|---|---|---|---|---|---|---|---|
| | Adjacency | | | Orientation | | | Causal Accuracy |
| | Recall | Prec | F1 | Recall | Prec | F1 | |
| *FCause* | **0.04** | **0.10** | **0.08** | **0.10** | **0.13** | **0.13** | **0.10** |
| *Notears-MLP* | 0.25 | 0.12 | 0.11 | 0.29 | 0.23 | 0.18 | 0.30 |
| *Notears-Sob* | 0.32 | 0.51 | 0.14 | 0.33 | 0.51 | 0.19 | 0.33 |

Table 3: **The relative difference between the performance of *FCause* with and without standardized data is lowest across all metrics.** Best results in bold. The table shows, for each metric, the relative difference between the performances achieved by each method with and without data standardization (lower is better). The table reports mean values across datasets.

The same *cannot* be said for *Notears-MLP* and *Notears-Sob*, whose performances vary (sometimes significantly) depending on whether the data is standardized or not.

To quantify this variation, table 3 shows the relative difference between the performances achieved by each method with and without standardizing the data. For each method, metric and dataset, the relative difference between the performance obtained with and without data standardization is computed as $RD(a,b) = 2|a - b|/(a + b)$. The table reports mean values across datasets. It can be observed that *FCause* achieves the lowest relative difference across all metrics by a large margin, followed by *Notears-MLP*, and finally by *Notears-Sob*.

Finally, we note that *FCause* can be used for missing value imputation, using the mean of $q_\psi(x_u|x_o)$ to impute missing values. We show imputation results in appendix H, where it can be observed that *FCause* performs similarly to *mice* (Van Buuren & Groothuis-Oudshoorn, 2011) and *missforest* (Stekhoven & Bühlmann, 2012), and outperforms *PVAE* (Ma et al., 2018).

## 7    DISCUSSION AND FUTURE WORK

We proposed *FCause*, a scalable causal discovery method able to model complex nonlinear relationships between variables that can handle datasets with missing values. Our results indicate that *FCause* could be impactful in real-world applications, where the functional form for the underlying model is not known, data normalization is out of the practitioner's control, and there may be missing values in the datasets. *FCause* has shown to yield good results in all these scenarios.

As future work, we identify two possible paths for further improvement. One involves the use of more powerful density approximators to model the noise variables. While we use a Gaussian with learnable scales, other options, such as one dimensional flows, could be tried. The other improvement involves extending the method to models more general than ANMs, such as the post nonlinear (Zhang & Hyvarinen, 2012) or affine (Khemakhem et al., 2021). These extensions should be possible, since all transformations involves in these kinds of models are invertible. Interestingly, once developed, these extensions could be readily applied to other causal discovery methods based on continuous optimization, thanks to the unifying framework presented in section 5.

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

# A  OPTIMIZATION DETAILS

As mentioned in the main text, we gradually increase the values of $\rho$ and $\alpha$ as optimization proceeds, so that non DAGs are heavily penalized. Inspired by *Notears*, we do this with a method that resembles the updates used by the augmented Lagrangian procedure for optimization (Nemirovsky, 1999). The optimization process interleaves two steps: (i) Optimize the objective for fixed values of $\rho$ and $\alpha$ for a certain number of steps; and (ii) Update the values of the penalty parameters $\rho$ and $\alpha$. The whole optimization process involves running the sequence (i)-(ii) until convergence, or until the maximum allowed number of optimization steps is reached.

**Step (i).** Optimizing the objective for some fixed values of $\rho$ and $\alpha$ using Adam (Kingma & Ba, 2014). We optimize the objective for a maximum of 500 epochs or until convergence, whichever happens first (we assume convergence if the loss does not improve during 100 epochs. If so, we move to step (ii)). We use Adam, initialized with a step-size of 0.01. During training, we reduce the step-size by a factor of 10 if the training loss does not improve for 50 steps. We do this a maximum of two times. If we reach the condition a third time, we do not decrease the step-size and assume optimization converged, and move to step (ii).

**Iterating (i)-(ii).** We initialize $\rho = 1$ and $\alpha = 0$. At the beginning of step (i) we measure the dag-penalty $P_1 = \mathbb{E}_{q_\phi(A)} h(A)$. Then, we run step (i) as explained above. At the beginning of step (ii) we measure the dag-penalty again, $P_2 = \mathbb{E}_{q_\phi(A)} h(A)$. If $P_2 < 0.65\, P_1$, we leave $\rho$ unchanged and update $\alpha \leftarrow \alpha + \rho\, P_2$. Otherwise, if $P_2 \geq 0.65\, P_1$, we leave $\alpha$ unchanged and update $\rho \leftarrow 10\, \rho$. We repeat the sequence (i)-(ii) for a maximum of 25 steps or until convergence (measured as $\alpha$ or $\rho$ reaching some max value), whichever happens first. The approach is summarized in algorithms 1 and 2.

**Other details.** We use $\lambda_s = 5$ (the regularization factor favouring sparsity).

---

**Algorithm 1** Outer training loop (iterating (i) and (ii))

---

**Require:** Sparsity factor $\lambda_s$, temperature for Gumbel-softmax $\tau$, dataset $D = \{x^n\}_{n=1}^N$, maximum
  values $\rho_{\max}, \alpha_{\max}$
  Initialize $(\rho, \alpha) = (1, 0)$
  Initialize each component of $q_\phi(A)$ to a Bernoulli with $p = 0.1$
  Initialize $N_{\text{iters}} = 0$
  **while** $\rho < \rho_{\max}$ & $\alpha < \alpha_{\max}$ & $N_{\text{iters}} < 25$ **do**
    Estimate DAG-penalty $P_1 = \frac{1}{M} \sum_{m=1}^M h(A_m)$, with $A_i \sim q_\phi(A)$
    Run inner optimization loop for $\alpha$ and $\rho$                    ▷ See algorithm 2
    Estimate DAG-penalty $P_2 = \frac{1}{M} \sum_{m=1}^M h(A_m)$, with $A_i \sim q_\phi(A)$
    **if** $P_2 < 0.65 \times P - 1$ **then**
      $\alpha = \alpha + \rho P_2$
    **else** $N$ is odd
      $\rho = 10 \times \rho$
    **end if** $N_{\text{iters}} = N_{\text{iters}} + 1$
  **end while**

---

# B  ELBO DERIVATION

The goal of maximum likelihood involves maximizing the likelihood of the observed variables. For *FCause* (with fully observed datasets) this corresponds to the log-marginal likelihood

$$\log p_\theta(x^1, \dots, x^N) = \log \sum_A p(A) \prod_n p_\theta(x^n | A). \tag{14}$$

Computing the marginalization from the equation above is intractable, even for moderately low dimensions, since the number of terms in the sum grows exponentially with the size of $A$ (which grows quadratically with the problem's dimensionality).

---

**Algorithm 2** Inner training loop (step (i))

---

**Require:** Sparsity factor $\lambda_s$, temperature for Gumbel-softmax $\tau$, dataset $D = \{x^n\}_{n=1}^N$
  Initialize $N_{\text{epochs}} = 0$
  Initialize step-size $\eta = 0.01$
  **while** Not converged & $N_{\text{epochs}} < 500$ **do**
    **for** $X_B \in D$ **do**                                                ▷ Batch of size 100
        Sample $A \sim q_\phi(A)$ with gumbel-softmax trick with temperature $\tau$
        Compute ELBO from eq. (6) using sample $A$ and $f_A(\cdot; \theta)$ for each $x \in X_B$
        Update $\phi$ and $\theta$ using $\nabla_{\phi,\theta}$ELBO with Adam update and step-size $\eta$
        **if** Condition to reduce step-size **then**
            $\eta = \eta/10$
        **end if**
    **end for**
    $N_{\text{epochs}} = N_{\text{epochs}} + 1$
  **end while**

---

Variational inference proposes to use a distribution $q_\phi(A)$ to build the ELBO, a lower bound of the objective from eq. (14), as follows:

$$\log p_\theta(x^1, \ldots, x^N) = \log \sum_A p(A) \prod_n p_\theta(x^n | A) \tag{15}$$

$$= \log \sum_A q_\phi(A) \frac{p(A) \prod_n p_\theta(x^n | A)}{q_\phi(A)} \tag{16}$$

$$= \log \mathbb{E}_{q_\phi(A)} \left[ \frac{p(A) \prod_n p_\theta(x^n | A)}{q_\phi(A)} \right] \tag{17}$$

$$\geq \mathbb{E}_{q_\phi(A)} \left[ \log \frac{p(A) \prod_n p_\theta(x^n | A)}{q_\phi(A)} \right] \qquad \text{(Jensen's inequality)} \tag{18}$$

$$= \mathbb{E}_{q_\phi(A)} \left[ \log p(A) \prod_n p_\theta(x^n | A) \right] + H(q_\phi) \tag{19}$$

$$= \text{ELBO}(\phi, \theta), \tag{20}$$

where we used that $H(q_\phi) = -\mathbb{E}_{q_\phi(A)} \log q_\phi(A)$ is the entropy of the distribution $q_\phi$. Interestingly, the distribution $q_\phi$ that maximizes the ELBO is exactly the one that minimizes the KL-divergence between the approximation and the true posterior, $\text{KL}(q_\phi(A) \| p_\theta(A | x^1 \ldots, x^N))$ (see, e.g., (Blei et al., 2017)). This is why $q_\phi$ can be used as a posterior approximation.

## C  DETAILS ON THE UNIFIED FLOW-BASED FORMULATION

**Details on weighted adjacency matrices**  These are used by *Notears*, *Notears-MLP*, *Notears-Sob* and *Grandag*, among other methods. We will present the formulation from (Zheng et al., 2020) since it is the more general one. Simply put, given a function $f(x; \theta) : \mathbb{R}^d \to \mathbb{R}^d$, Zheng et al. (2020) propose to compute each entry of the weighted adjacency $W(\theta)$ as

$$W(\theta)_{j,i} = \|\partial_j f_i\|_{L^2}, \tag{21}$$

where $\partial_j f_i$ denotes the partial derivative of $f_i(x; \theta)$ with respect to $x_j$. Clearly, if the output of $f_i(x; \theta)$ does not depend on the value of $x_j$, then we get $W(\theta)_{j,i} = 0$.

While this is a useful general notion to build these weighted adjacency matrices, it is not easy to apply in practice. Zheng et al. (2020) propose some specific alternatives for a few cases, including the case where each $f_i$ is given by a multi layer perceptron (MLP) , or as a weighted combination of nonlinear basis functions. We include the details for MLPs here. In this case, each $f_i$ is given by an independent MLP. Let $Q_i^{(1)}$ represent the weights of the first layer of $f_i$. If the $j$-th column of $Q_i^{(1)}$ is all zeros, then the output of $f_1$ cannot depend on $x_j$. Using this insight Zheng et al. (2020)

propose to build the weighted adjacency matrix as

$$W(\theta)_{j,i} = \|j^{th}\text{column}(Q_i^{(1)})\|_{L^2}. \tag{22}$$

**Details for the *Notears* case**  We show full details for the flow formulation of (linear) *Notears*. The details for the other methods follow this closely. *Notears* (linear) finds the optimal weighted adjacency $W$ by solving

$$\min_{W \in \mathbb{R}^{d \times d}} \sum_{n=1}^{N} \|x^n - W x^n\|^2 \qquad \text{s.t.} \qquad h(W) = 0, \tag{23}$$

where $h(W) = 0$ exactly characterizes DAGs (eq. (3)). As proposition 1 states, the exact same optimization problem is obtained when fitting a flow under acyclicity constraints.

**Proposition 1.** *Fitting a flow using a standard Gaussian as base distribution $p_z$, the transformation $z = x - Wx$, and a hard constraint enforcing acyclicity on $W$ results in exactly the same optimization problem as the one in eq. (23) used by Notears (Zheng et al., 2018).*

*Proof.* We enforce the acyclicity constraint on $W$ by restricting the optimization domain to $\Omega = \{W : h(W) = 0\}$. Then, the maximum likelihood objective can be written as

$$\sum_n \log p(x^n) = \sum_n \log p_z(x^n - Wx^n) + \log|\det(I - W)| \qquad \text{(change of variable)} \tag{24}$$

$$= \sum_n \log p_z(x^n - Wx^n) \qquad \text{(lemma 1)} \tag{25}$$

$$= -\sum_n \|x^n - Wx^n\|^2 \qquad (p_z \text{ is std Gaussian}), \tag{26}$$

where eq. (24) is valid because the transformation $z = g(x) = x - Wx$ is invertible for any $W \in \Omega$. We see that we recover exactly the *Notears* objective from eq. (23). $\square$

## D  PROOF OF LEMMA 1

We split the proof in several simple steps.

1. $g_A(x) = x - f_A(x)$ has Jacobian-determinant $\det(I - J_A(x))$, where $J_A(x)$ is the Jacobian of $f_A(x)$.
2. $J_A(x)$ has non-zero entries exactly in the positions where $A^\top$ is non-zero. Therefore, it retains the DAG structure.
3. Matrices with a DAG structure are nilpotent (i.e., all eigenvalues are zero). Thus, $J_A(x)$ can be factorized as $J_A(x) = QUQ^*$, where $Q$ is unitary and $U$ is *strictly* upper triangular (Schur factorization).
4. Finally, $\det(I - f_A(x)) = \det(I - QUQ^*) = \det(I - U) = 1$.

## E  DAGS AND INVERTIBILITY

As mentioned in section 4, if $A$ is not a DAG the transformation $z = g_A(x) = x - f_A(x)$ may not be invertible. A simple example suffices to show this. Consider the two-variable scenario with

$$A = \begin{bmatrix} 0 & 1 \\ 1 & 0 \end{bmatrix}$$

and $f_A(x) = [f_1(x_2), f_2(x_1)]$, where $f_1(x_2) = x_2$ and $f_2(x_1) = x_1$. Then, computing $z = x - f_A(x)$ we get

$$z_1 = x_1 - x_2$$
$$z_2 = x_2 - x_1.$$

Clearly, the function $z = g_A(x) = x - f_A(x)$ is not invertible, since for any pair $(x_1, x_2)$ satisfying $x_1 = x_2$ we get $z_1 = z_2 = 0$.

## F    RESULTS WITH HIGHER LEVEL OF NOISE

This section shows results in the exact same setting as the one described in section 6, with the difference being that synthetic datasets are generated with a higher level of noise (the real and pseudo-real datasets are exactly the same as before). This is achieved by sampling the noise scale $\sigma_i \sim \mathcal{U}(0.2, 2)$ (instead of $\sigma_i \sim \mathcal{U}(0.2, 1)$). Results are shown in fig. 2. As expected, as a consequence of the higher level of noise, all methods perform slightly worse. However, the main conclusion remains unchanged, *FCause* outperforms other methods and is still competitive even when ran on the datasets with 30% of missing values.

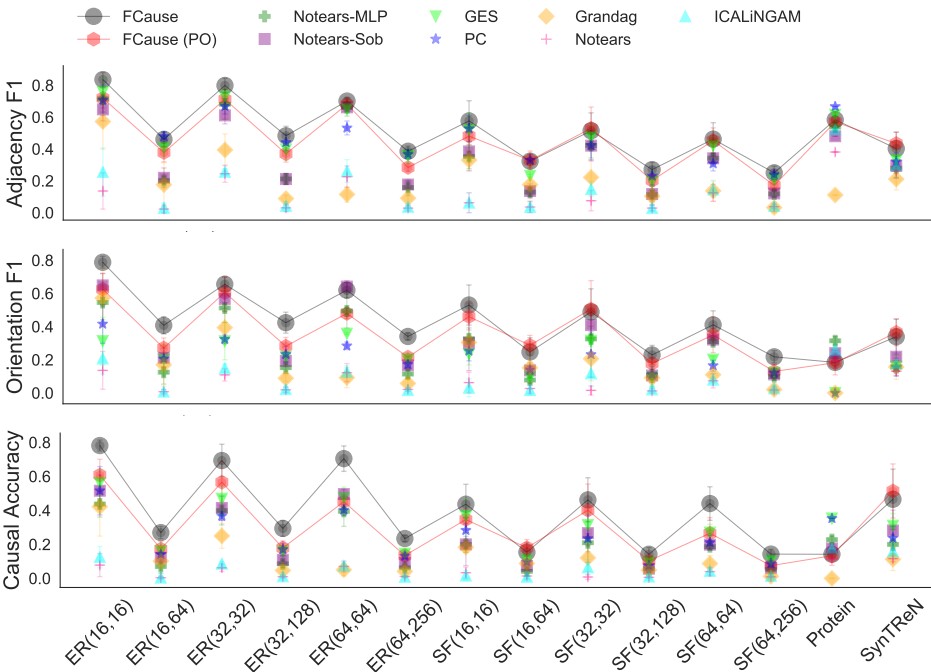

Figure 2: ***FCause* achieves better results than the baselines in all three metrics shown.** Results obtained with the dataset with higher levels of noise. The legend "FlowVICause (PO)" corresponds to running *FCause* with 30% of the training data missing completely at random. For readability, we highlight the *FCause* and *FCause (PO)* results by connecting them with soft lines. For clarity the figure shows mean results across four different random seeds.

## G    FULL RESULTS WITH STANDARD DEVIATIONS

Results for all metrics, including standard deviations, are shown in fig. 3. It can be observed that some methods achieve higher precision values than *FCause* for some datasets. Looking at the results in detail, this is not surprising. The methods that achieve better precision values always have extremely low recall. Meaning that they detect very few edges, which tend to be correct. This leads to quite high precision values, but very low recall, F1 and causal accuracy. That is, their overall performance is quite poor, since they fail to detect most edges. (e.g. Suppose that a problem has 100 edges. A method that detects just one edge, and that edge is correct, gets a precision of one, but a recall and F1 of almost zero). On the other hand, while *FCause* tends to achieve slightly lower precision values, its overall performance is significantly better.

## H    MISSING VALUES IMPUTATION

**Datasets.** We use the same datasets described in section 6, all standardized. We remove 30% of the values in the training set randomly. The testing sets use are composed of $n = 500$ samples for the

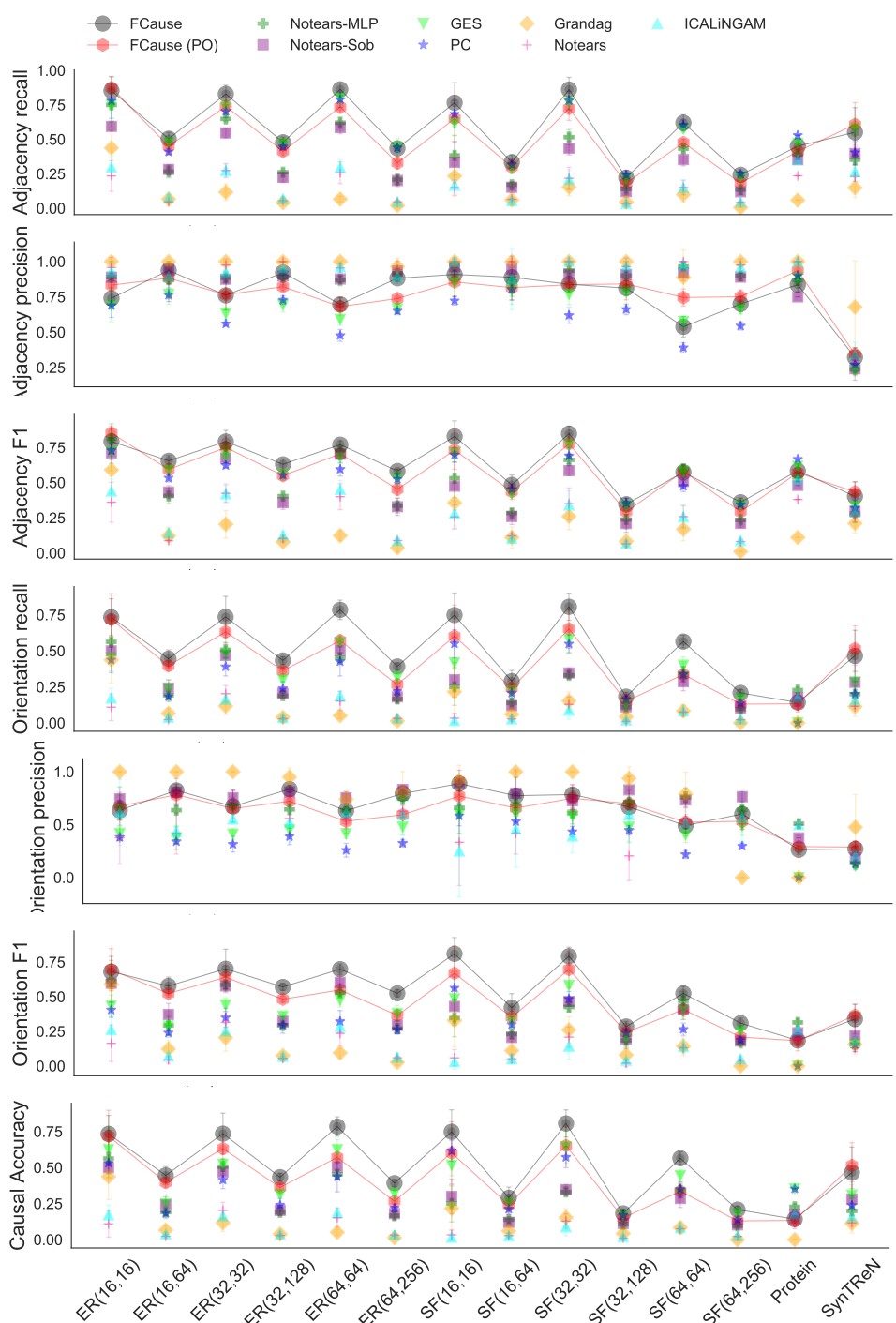

Figure 3: Results for all metrics including standard deviations computed over runs with four different random seeds. For readability, we highlight the *FCause* and *FCause (PO)* results by connecting them with soft lines.

synthetic datasets, $n = 100$ samples for the pseudo-real data obtained with the SynTReN generator, and $n = 53$ samples for the protein cells dataset.

**Baselines.** We compare *FCause* against *mean imputing*, *mice* (Van Buuren & Groothuis-Oudshoorn, 2011), *missforest* (Stekhoven & Bühlmann, 2012) and *PVAE* (Ma et al., 2018). We also include as baseline *FCause* with an adjacency matrix fixed to a strictly upper triangular. We call this method *UTFlow*.

**Metrics.** We report the normalized Root Mean Squared Error (NRMSE) computed on the test set. All results are computed as the mean across four different random seeds.

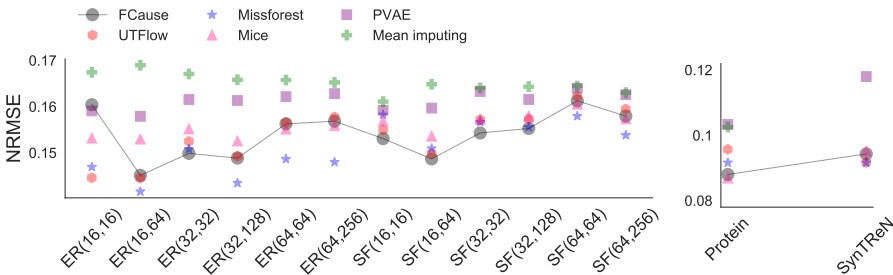

Figure 4: Imputation results. All results are obtained as the mean over four runs with different random seeds. For readability, we highlight the *FCause* results by connecting them with soft lines. For SynTRen, mean imputing achieves a NRMSE of 0.29, not shown in the plot.

Results are shown in fig. 4. For clarity, all values reported are the mean across four runs with different random seeds. There is no single method that dominates across all datasets. Overall, it can be observed that *Missforest* achieves the best performance, followed by *FCause* and *UTFlow*, and finally *Mice*. *PVAE*, while better than *mean imputing*, performs worse than other methods in most datasets. Perhaps surprisingly, *UTFlow* and *FCause* achieve very similar performances.

