# OpenReview forum: "FCause: Flow-based Causal Discovery"
_ICLR.cc/2022/Conference — ICLR 2022 Submitted_

### Official Review · Reviewer_q8GH · 2021-10-17

**Correctness:** 2
**Technical Novelty And Significance:** 2
**Empirical Novelty And Significance:** 2
**Recommendation:** 3
**Confidence:** 3

**Main Review:**

Strengths:

•	First paper to combine VB with NOTEARS

•	Empirically, it does well in simulations

•	It can handle missing values and allows nonlinear functional relationships

Weaknesses:

•	The motivation of using DAG is unclear. DAG is a special case of directed cyclic graphs (DCG). One important advantage of DAG over DCG is its simple factorization, which leads to more efficient computation than DCG. With the NOTEARS framework, this advantage seems largely lost because during the training, at least in the early phase, the graphs may be not be acyclic. Then why one even makes an often unrealistic assumption of acyclicity to begin with? In a typical run of the proposed algorithm, is the DAG factorization ever exploited? If it is exploited in iterations when the graph is acyclic, then typically in how many iterations the graph is acyclic? In summary, acyclicity is, in my opinion, a modeling restriction, not a feature. Is the proposed algorithm faster than the same algorithm without the h(A) penalty (which is both simpler computationally and more general in terms of modeling)?

•	The proposed approach seems doing very well in various simulations but doesn’t do well in the real data, which questions the real practical advantage of the proposed method. Perhaps including more real data analyses/comparison can help support it.

•	It would be helpful for readers if computation/algorithm/implementation details are given.

•	Not clear why Carefl is relevant to this paper.  Maybe I missed something, but the only place Carefl is mentioned in the method description is around Equation (6). But it looks to me just a standard definition of a nonlinear SEM? See e.g., Equation (4) in “On Causal Discovery with Cyclic Additive Noise Models”.

•	Scalability is claimed but not tested: the largest dimension investigated is 64.

•	Although the proposed method can handle missing data, missing at random is a rather strong assumption. Under this assumption, practically any Bayesian methods can easily impute the missing values.

•	Is Lemma 1 a well-known result for any DAG? The Jacobian can be transformed to an upper or lower triangular matrix with unit diagonals by permuting the nodes and hence the determinant is trivially 1. I might have missed something here.

•	What is the rationale of imposing prior on A but not on theta. This choice loses some uncertainty in addition to that lost by using Variational Bayes.

**Summary Of The Paper:**

The paper proposes a flow-based causal discovery method to learn nonlinear causal DAGs with potential missing values. The proposed approach is based on variational Bayes and continuous optimization approach.

**Summary Of The Review:**

In summary, while the paper has some interesting idea of combining variational Bayes algorithm with NOTEARS (continuous optimization) for nonlinear DAG learning, the significance of the contribution is not yet strongly supported theoretically or empirically.

---

> ### Author Response · Authors · 2021-11-22
> **Thank you for reviewing our work**
>
> **The motivation of using DAG is unclear. DAG is a special case of directed cyclic graphs (DCG) [...] Is the proposed algorithm faster than the same algorithm without the h(A) penalty (which is both simpler computationally and more general in terms of modeling)?**
>
> The DAG assumption is likely the most common one in the field of causal discovery. While there are a few papers that consider cyclic graphs (e.g. [B, C]), we do not consider the use of this assumption as a big drawback of the paper. In fact, dealing with cycles becomes quite problematic for this type of models. This is because it is not easy to ensure invertibility of the transformation used. For instance, the paper "On Causal Discovery with Cyclic Additive Noise Models" discusses this. In the "Interpretation in the cyclic case" section/paragraph they talk about the number of fixed points of the equation $x = f(x) + e$, which in general may be some number between 0 and $\infty$. To ensure invertibility the number of fixed points must be exactly one. Ensuring this is not straightforward; Lemma 1 in their paper gives a sufficient (but not necessary) condition for the bivariate case (involving the derivatives of the maps), which is quite restrictive (in fact, in Appendix E we show a simple example that does not satisfy that condition and leads to a non-invertible transformation). Also, when dealing with cyclic graphs the log-det Jacobian term cannot be ignored. This may or may not be an issue depending on the model, which determines  the cost of computing it (typically cubic in dimension, unless special structure is available).
>
> The fact that invertibility conditions are problematic is acknowledged (in a possibly more indirect way) in other papers too. For instance [A] comments that the factorization they use (equation 10 in their paper) is only valid when the graph is a DAG (right after equation 10).
>
> [A] Gradient based neural dag learning (Lachapelle et al, 2020).
>
> [B] On causal discovery with cyclic additive noise model (Mooij et al, 2011).
>
> [C] Causal calculus in the presence of cycles, latent confounders and selection bias (Forre et al 2020).
>
> **It would be helpful for readers if computation/algorithm/implementation details are given.**
>
> We added details in Section 6 and Appendix A.
>
> **Not clear why Carefl is relevant to this paper [...]**
>
> We took Carefl, which formalized the connection between autoregressive flows and causality, as the starting point for our method. However, as other reviewers also noticed, work prior to Carefl proposed related methods. We edited the related work section to reflect this.
>
> **Scalability is claimed but not tested: the largest dimension investigated is 64.**
>
> See general answer.
>
> **Although the proposed method can handle missing data, missing at random is a rather strong assumption. Under this assumption, practically any Bayesian methods can easily impute the missing values.**
>
> The goal is to perform causal discovery in the presence of missing values efficiently. (Traditional deletion based method cannot handle such scenario efficiently, and will fail if no fully observed data is available). Our approach is one way to do this, using the learned causal model to guide inference over missing values (by approximating its posterior distribution). We consider extending the approach to the case where data is missing not at random to be an interesting direction.
>
> **Is Lemma 1 a well-known result for any DAG? The Jacobian can be transformed to an upper or lower triangular matrix with unit diagonals by permuting the nodes and hence the determinant is trivially 1. I might have missed something here.**
>
> The result is simple. After submission we noticed that it is stated in previous work, we changed our presentation to reflect this.
>
> **What is the rationale of imposing prior on A but not on theta.**
>
> The parameters theta could be treated as random variables too. This is a modelling choice. However, treating them as parameters simplifies the method significantly, as performing inference reliably over the parameters of neural networks is known to be hard. We do not see this choice as a negative aspect of our method. (Additionally, this choice is quite popular. It is used, for instance, with VAEs, were the model parameters are not treated as random variables.)

---

### Official Review · Reviewer_m3am · 2021-10-31

**Correctness:** 3
**Technical Novelty And Significance:** 3
**Empirical Novelty And Significance:** 3
**Recommendation:** 6
**Confidence:** 4

**Main Review:**

### Pros
- The paper is well written and easy to follow.
- The paper develops a unified fiew of existing continous optimization methods for learning DAGs, which provide insights into how those methods relate to one another.

### Cons
- In terms of methodology, the idea of normalizing flow has been explored by [1]. I would encourage the authors to include explanation about the difference between their work and [1].
- Also, the methods of using gumbel-softmax and binary matrix have been explored by [1, 2], which may be worth mentioning and comparing to.
- Regarding the experiments, it would be better to include more nonlinear/nonparametric baselines, such as GES with generalized score [3], PC with kernel-based test [4], DAG-GNN [5], CAM [6], and [2].
- The results of GraN-DAG seem to be much worse than those reported in the original paper. Did the authors use the preliminary neighborhood selection and pruning steps (Appendix A.3 in [7])?
- It would be better to include the structural intervention distance [8] that measures the accuracy of interventional queries and could be more informative.
- The proposed $f_i(x)$ with graph neural network seems to be restrictive, since it assumes a causal additive model (i.e. elementwise nonlinear relationship $g_j$) with a "post-nonlinear" operation $h_i$. However, it is surprising that this function works well with the Gaussian process dataset considered in the experiments, although the functional assumption is not met. Could the authors provide explanation on this?

### Other comments
- $h$ is used for different purposes, i.e. neural network and algebraic characterization of DAGs.
- I am not sure if it is appropriate to claim about causality without further conditions. As the authors know, previous works [5, 7, 9] focus on learning DAGs instead of claiming about causal discovery. Perhaps could the authors include some explanation on this?
- The graph neural networks used in Section 4.1 seem to be similar to [5, 10] which fall under the proposed unfied view; the authors could include comparison with them in Section 5.

### References
1. Differentiable Causal Discovery from Interventional Data, 2020.
2. Masked Gradient-Based Causal Structure Learning, 2020.
3. Generalized Score Functions for Causal Discovery, 2018.
4. Kernel-based Conditional Independence Test and Application in Causal Discovery, 2011.
5. DAG-GNN: DAG Structure Learning with Graph Neural Networks, 2019.
6. CAM: Causal additive models, high-dimensional order search and penalized regression, 2014.
7. Gradient-Based Neural DAG Learning, 2020.
8. Structural Intervention Distance (SID) for Evaluating Causal Graphs, 2015.
9. DAGs with NO TEARS: Continuous Optimization for Structure Learning, 2018.
10. A Graph Autoencoder Approach to Causal Structure Learning, 2019.

**Summary Of The Paper:**

This paper proposes a general flow-based approach to learn DAGs from data which provides a unified view of existing continuous optimization methods for structure learning. As a side benefit, the authors demonstrate that the proposed method could naturally be modified to handle missing data. The authors provide empirical studies to show that their proposed method outperforms the other baselines.

**Summary Of The Review:**

The paper develops a unified fiew of existing continous optimization methods for learning DAGs, which provide insights into how those methods relate to one another. However, I have some concerns about the experiments, and am willing to increase my score if the authors could address my comments.

---

> ### Author Response · Authors · 2021-11-22
> **Thank you for reviewing our work**
>
> **The methods of using gumbel-softmax and binary matrix have been explored by [1, 2], which may be worth mentioning and comparing to.**
>
> We added citations to these papers mentioning they also learn a binary adjacency matrix using the Gumbel-softmax. Just to clarify, we did not intend to claim this aspect of our work was novel, as the Gumbel-softmax trick is likely one of the most popular approaches to deal with discrete variables. Our contribution is a flow based method that can discover causal relationship in multivariate cases and a general framework for such type of methods.
>
> **Did the authors use the preliminary neighborhood selection and pruning (grandag)?**
>
> We do not use the preliminary neighborhood search. While it has been observed to be useful, this pre-processing step can be applied with several of the methods tested, so we decided to turn it off for the comparison (we added a note on this in the paper). We use the Jacobian-based pruning step.
>
> **It would be better to include the structural intervention distance.**
>
> While metrics other than the ones we use in the paper can be reported, we believe that the eight ones we use are informative regarding causal discovery. They are also quite standard, and used in several papers, including Glymour et al (2019), Tu et al (2019), Claassen and Heskes (2012), among others.
>
> **The proposed $f_i(x)$ with graph neural network seems restrictive [...]**
>
> The formulation is not restrictive, thanks to the fact that $g_j$ outputs a vector of dimension $d$ (if the output was a scalar then it would be). To see that a vector of dimensionality $d$ is enough, it is sufficient to note that $g_j(x_j)$ can potentially output a vector of dimension $d$ that's all zeros except it's $j$-th component, which takes the value $x_j$ (and thus, the input to $f_i$ would be a vector containing exactly the values of the parents of $x_i$).
>
> **I am not sure if it is appropriate to claim about causality without further conditions. As the authors know, previous works [5, 7, 9] focus on learning DAGs instead of claiming about causal discovery. Perhaps could the authors include some explanation on this?**
>
> First, we believe that “structure learning”, “causal discovery” and “DAG learning” have been used in a somewhat exchangeable way in the literature. As pointed out by [A], while some papers claim to be learning a DAG, in the community, they have been widely considered to be causal discovery methods. Also, many of the original DAG learning papers are evaluated in several causal discovery benchmarks.
> Secondly, there has been some dabate about the nature of causality (see, e.g., section A3 of [B]). However, such debating mainly considered the dynamical system perspective, which is out of our scope.
> Finally, as discussed in the “identifiable” part of the answer (see above), we follow the traditional causal discovery convention using maximum likelihood estimation to discovery the underlying causal graph. (There are strong theoretical results relating this to score-based methods and information-based approaches [C, D]).
>
> [A] Unsuitability of NOTEARS for Causal Graph Discovery (Marcus Kaiser and Maksim Sipos, 2021).
>
> [B] Graphical modelling in continuous-time: consistency guarantees and algorithms using Neural ODEs (Bellot et al, 2021).
>
> [C] On Estimation of Functional Causal Models: General Results and Application to the Post-Nonlinear Causal Model (Zhang et al, 2015).
>
> [D] Causality Discovery with Additive Disturbances: An Information-Theoretical Perspective (Zhang et al, 2009).
>
> **The graph neural networks used in Section 4.1 seem to be similar to [5, 10] which fall under the proposed unified view; the authors could include comparison with them in Section 5.**
>
> The approach by [5] (DAG-GNN), while related, does not really fit the unified view as presented. This is because in that work the relation between the noise $z$ and the observations $x$ is not invertible. Thus, their approach does fall within the group of flow-based methods (it is presented as a VAE, which allows non-invertible functions by optimizing a lower bound instead of the true marginal likelihood). The proposed unified view could be extended to include such a case, but as presented it does not.
>
> On the other hand, the method from [10] does indeed fit the unified view. We added it to Section 5. Similarly to Notears, our unified view shows that the method from [10] can be formulated from a probabilistic perspective using autoregressive flows.

---

> > ### Comment · Reviewer_m3am · 2021-11-23
> > **Thank you for the response**
> >
> > Thank you for the detailed response. Most of my concerns have been addressed, and I have increased my score to 6. However, my concern regarding the performance of GraN-DAG persists. In particular, GraN-DAG adopts a preliminary neighborhood step and two different pruning steps, i.e., the pruning step identical to CAM to remove spurious edges, and the Jacobian-based pruning step to ensure acyclicity. If I understand correctly, in the paper, the authors only adopted the Jacobian-based pruning step. While it is true that the preliminary neighborhood selection and CAM-based pruning step can also be applied with several of the other baselines considered, these two steps have particularly large impacts on the performance of GraN-DAG, as compared to the other baselines such as NOTEARS; see Table 5 and 6 in the GraN-DAG paper. That is, without these two additional steps, the performance of GraN-DAG will be very bad, while adding these two steps may not improve the other baselines much. Therefore, I think it would be better to conduct an additional empirical study that includes these two steps for GraN-DAG (or even, all other methods).

---

> > > ### Author Response · Authors · 2021-11-24
> > > **Thank you**
> > >
> > > We thank you for increasing the score considering our response.
> > > We agree that it will be interesting to add experiments with preliminary neighborhood selection for all methods, compare their performance in such a setting. We will add it to the revised version of the paper.

---

> > > > ### Comment · Reviewer_m3am · 2021-11-24
> > > > **Preliminary neighborhood search and CAM-based pruning**
> > > >
> > > > Thank you. It will be interesting to add experiments for both the preliminary neighborhood search and the CAM-based pruning step.

---

### Official Review · Reviewer_Du7p · 2021-11-02

**Correctness:** 3
**Technical Novelty And Significance:** 3
**Empirical Novelty And Significance:** Not applicable
**Recommendation:** 5
**Confidence:** 3

**Main Review:**

The authors consider a fundamental problem of learning directed causal graphs with nonlinear continuous data.  They propose a scale causal discovery method based on the flow, variational inference, GNN, and Notears.  Their method can handle high dimensional and missing data.

However, there are some concerns / suggestions:

1. It's not clear to me whether the proposed optimization formulation (Equation 6 or 10) can always render the highest score? I notice that it does not restrict the distribution of noise z. Is there any theorem to support it? Please clarify it.
2. I strongly recommend that the authors should give a summary, such as an algorithm, to show the optimization process.
Regarding the unified flow-based formulation (lemma 2).
3. As far as I know, in the additive noise setting, the log-likelihood of observed variables can be converted into the log-likelihood over the noise estimations, such as in the following paper. This is not surprising to me though the author provides another strategy to prove it.

Cai R, Qiao J, Zhang Z, et al. Self: structural equational likelihood framework for causal discovery[C]//Thirty-second AAAI conference on artificial intelligence. 2018.

4.  The authors should explain why the metrics of the other methods cannot perform well. For example,  Notear perform incredibly bad, why?
5. More experiments are needed, report based on only four data sets is unconvincing.

minor typos:

Equation 3: no symbol description for tr,e.


If the authors sufficiently address the mentioned concerns, I am happy to change my assessment.

**Summary Of The Paper:**

This  paper  focuses on causal discovery,  especially  on learning DAG using continuous optimization methods. \
In the nonlinear additive noise setting, the authors propose a flow-based method, called FCause, to optimize the log-likelihood of the observed variables.

**Summary Of The Review:**

The authors address an important and hard problem in learning DAG. The method is a novel and interesting contribution to the continuous optimization literature

---

> ### Author Response · Authors · 2021-11-22
> **Thanks you for reviewing our work**
>
> **It's not clear to me whether the proposed optimization formulation (Equation 6 or 10) can always render the highest score? [...]**
>
> See identifiablility discussion in the general answer. In addition, by [A] the only restriction needed for identifiability in nonlinear additive noise models is the independence of the noise variables, which holds in the model we propose. Specifically, we set the noise variables to be Gaussian with learnable variance, but this can be easily extended to any other choice as long as the noise variables are independent (could use, for instance, an independent 1D flow for each noise variable).
>
> [A] Nonlinear causal discovery with additive noise models (Hoyer et al, 2008).
>
> **I strongly recommend that the authors should give a summary, such as an algorithm, to show the optimization process.**
>
> We included an algorithm in Appendix A.
>
> **As far as I know, in the additive noise setting, the log-likelihood of observed variables can be converted into the log-likelihood over the noise estimations, such as in the following paper (Self: structural equational likelihood framework for causal discovery). This is not surprising to me though the author provides another strategy to prove it.**
>
> We took the carefl framework which formalizes the connection to autoregressive flows as basis of our approach. We agree that the paper mentioned (and others) present closely related ideas. We added this reference to the related work section to reflect this.
>
> **The authors should explain why the metrics of the other methods cannot perform well. For example, Notears perform incredibly bad, why?**
>
> Notears can only model linear relationships between variables. It is expected to perform poorly on the experiments tested. The synthetic data was generated using non-linear functions, and there's no reason to believe that the real and pseudo-real datasets satisfy linear relationships between variables. The nonlinear versions of notears have been observed to have some issues dealing with different scalings of the data (Reisach et al 2021, Kaiser and Sipos 2021), and in our simulations we obtained different results depending on whether the data used is standardized or not.

---

### Official Review · Reviewer_DG4z · 2021-11-02

**Correctness:** 4
**Technical Novelty And Significance:** 3
**Empirical Novelty And Significance:** Not applicable
**Recommendation:** 8
**Confidence:** 3

**Main Review:**

Strengths:

- The method presented in the paper is scalable and empirically gives good results in various settings.

- All aspects of the method are founded on theoretical developments in many earlier papers.

- The developments are also used to state a framework that makes it much easier to state the relations between different algorithms in prior work.

Concern:

- notears turned out to have a particular weakness to different noise variances. Could you discuss possible weaknesses that your proposed method might have?

Minor points:

- Final sentence in section 2, "better properties" - better than what?

- page 3: double "to"

- page 5: double "to"

- page 9: "practitioners" -> "practitioner's"

**Summary Of The Paper:**

This paper proposes to combine the continuous optimization-based causal discovery approach from notears with flow-based function learning. An extension is given to data missing (completely) at random. Also, a unifying framework is presented facilitating the comparison and exchange of ideas between different continuous optimization-based causal discovery methods, viewing all in terms of flows.

**Summary Of The Review:**

I think this is a strong paper, both for the main result as well as for the unifying framework in section 5. My only concern is that the proposed method has many "moving parts" making it hard to see how it might or might not succeed in particular situations. But the empirical results are very strong, so that I recommend acceptance.

---

> ### Author Response · Authors · 2021-11-22
> **Thank you for reviewing our work**
>
> **Could you discuss possible weaknesses that your proposed method might have?**
>
> Our method inherits some weaknesses of previous approaches for causal discovery based on continuous optimization. For instance, we assume causal sufficiency, which means that our method cannot handle unobserved confounders, which is common in real-world applications. Moreover, our method can only handle continuous variables at the current stage which is a limitation of flow base methods (though research on discrete flows is active.) Additionally, as presented the model assumes the ANM form (though can be expanded, see discussion and future work section), and does not handle values missing not at random.
>
> **My only concern is that the proposed method has many moving parts [...].**
>
> The method does involve a few choices, such as the variational distribution used for $A$ and to model missing values, the architecture of the amortization network for approximate inference when values are missing, noise distribution, etc. While we did not explore different alternatives thoroughly, we achieved strong results with fairly simple and standard choices: a factorized Bernoulli for $A$, a factorized Gaussian for the value imputation part, a simple two layer feed forward network for amortization, and we fix the noise distribution to a Gaussian. We believe more powerful choices may further improve results, and would be interesting to explore as future work.
>
> Similarly, it is worth mentioning that other methods also have some parameters. For instance, the $p$ value for PC, the choice of function approximations by non-liner notears (neural network architecture, basis expansion), etc.

---

> > ### Comment · Reviewer_DG4z · 2021-11-29
> > **Response read**
> >
> > Thank you for addressing my questions. It's good to know that standard choices sufficed to get your empirical results.
> >
> > Continuous optimization for structure learning is not something I'm working in myself, but I have noticed such methods getting strong results. I appreciate this paper's contribution to this promising line of work, and in particular its unifying framework that sheds light on the relation between such methods. I am still in favor of accepting this paper.

---

### Official Review · Reviewer_2kV5 · 2021-11-03

**Correctness:** 2
**Technical Novelty And Significance:** 2
**Empirical Novelty And Significance:** 2
**Recommendation:** 3
**Confidence:** 4

**Main Review:**

The paper builds up on Carefl (Khemakhem et al 2021), which fits an autoregressive flow model using the variables' causal ordering, extending it to use a flow-based model for inferring the causal structure.

One of the issues I have is that it is supposed to provide a more scalable alternative to better understood methods, e.g. constraint based methods like PC or FCI. Both of these methods can easily scale up to hundreds of variables (also since they do not exhaustively search over the space of all DAGs but work on equivalence classes) and provide theoretical guarantees. Moreover it does not provide any identifiability result, showing that the learnt graph is indeed causal. Given the approximated posterior, this cannot be directly derived from standard score-based methods.

**Summary Of The Paper:**

The paper proposes a flow-based causal discovery method that is supposed to be scalable (in the experiments d=16,32,64) and able to impute missing values.

**Summary Of The Review:**

The paper seems to be a relatively incremental contribution, that is not embedded in the related work on causal discovery and does not provide any theoretical guarantee that one would learn a causal graph.

---

> ### Author Response · Authors · 2021-11-22
> **Thank you for reviewing our work**
>
> We believe that we have addressed all your questions in the general answer. Please let us know if you have further questions.

---

### Author Response · Authors · 2021-11-22
**General concerns**

We thank all reviewers for the time. We address some of the questions that are raised by more than one reviewers in this section and address reviewer specific questions in the individual replies.

**Scalability and PC, FCI**

Traditional methods such as PC and FCI are indeed scalable. However, there are many limitations of such methods that limits their real-world application impact. To name a few: such constrained based methods rely on conditional independent test, which is difficult by itself [A] and very sample inefficient as shown in [B] where even with tens of thousands data-points the results are far away from practical usage; they return a CPDAG or PAG, which requires domain knowledge to interpret. Thus, such methods cannot satisfy the needs for major application usages in real-world.

For more advanced score based methods, such as [C, D, E, F], they indeed can have improved performance in terms of causal accuracy when compared to traditional methods, but suffer from poor scalability and may not be applicable even with less than 100 variables.

The further show our method's scalability, we ran simulations on synthetic data (ER graph) with dimensionality 128 and 256. We got the following results (we also run PC to compare results):
- Dimension 128
   - Fcause. Adjacency F1: 0.81, Orientation F1: 0.62, Causal accuracy: 0.68.
   - PC. Adjacency F1: 0.46, Orientation F1: 0.3, Causal accuracy: 0.5.
- Dimension 256
   - Fcause. Adjacency F1: 0.73, Orientation F1: 0.45, Causal accuracy: 0.49.
   - PC. Adjacency F1: 0.37, Orientation F1: 0.21, Causal accuracy: 0.46.


[A] Shah, Rajen D., and Jonas Peters. "The hardness of conditional independence testing and the generalised covariance measure." The Annals of Statistics 48.3 (2020): 1514-1538.

[B] Tu, Ruibo, et al. "Neuropathic Pain Diagnosis Simulator for Causal Discovery Algorithm Evaluation."

[C] Koivisto, Mikko, and Kismat Sood. Exact Bayesian structure discovery in Bayesian networks (2004).

[D] Ott, Sascha, Seiya Imoto, and Satoru Miyano. Finding optimal models for small gene networks (2004).

[E] Silander, Tomi, and Petri Myllymaki. A simple approach for finding the globally optimal Bayesian network structure (2012).

[F] Yuan, Changhe, and Brandon Malone. Learning optimal Bayesian networks: A shortest path perspective (2013).


**Concerns on identifiability**

In our work we deal with nonlinear additive noise models, whose causal graph is known to be identifiable [G]. Additionally, it has been shown that maximum likelihood can be used to recover the causal graph for identifiable models [H, I]. This is exactly what our method does: it attempts to maximize the likelihood by maximizing a tractable lower bound, the ELBO. Additionally, if the approximate posterior $q(A)$ equals the true posterior, then the bound is tight (i.e. the ELBO becomes exactly the likelihood).

[G] Nonlinear causal discovery with additive noise models (Hoyer et al, 2008).

[H] On Estimation of Functional Causal Models: General Results and Application to the Post-Nonlinear Causal Model (Zhang et al, 2015).

[I] Causality Discovery with Additive Disturbances: An Information-Theoretical Perspective (Zhang et al, 2009).

**More datasets and baselines**

We agree that adding more baselines would make the paper stronger. However, we believe that the baselines considered are representative enough to show our contributions. Regarding datasets, we followed [J], which use some synthetic data similar to ours, pseudo-real data with the Syntren generator, and real data from [K].

[J] Gradient based neural dag learning (Lachapelle et al, 2020).

[K] Causal protein-signaling networks derived from multiparameter single-cell data (Sachs et al, 2005).

---

### Author Response · Authors · 2021-11-29
**Thank you**

We thank reviewer DG4z  for your support again based on the strong perfromance of the proposed method and the contribution of unified view. Dear reviewer q8GH, Du7p and 2kV5, please let us know if you have any other question after reading our responce.

---

### Decision · Program_Chairs · 2022-01-20

**Decision:**

Reject

**Comment:**

This paper proposed a flow-based approach FCause to Bayesian causal discovery that is scalable, flexible, and adaptive to missing data. Reviewers were split on this paper and could not reach a consensus during the discussion, and no reviewer pushed for acceptance. After taking a closer look myself, I agree with several of the reviewers that while the core ideas here are interesting and novel, there remain too many unresolved issues that require another round of revision.

I encourage the authors to carefully take in account the reviewers' comments and re-submit this promising work to another ML venue.